# *In vitro* selection of Remdesivir resistance suggests evolutionary predictability of SARS-CoV-2

Agnieszka M. Szemiel[1☯], Andres Merits[2], Richard J. Orton[1], Oscar A. MacLean[1], Rute Maria Pinto[1], Arthur Wickenhagen[1], Gauthier Lieber[1¤], Matthew L. Turnbull[1], Sainan Wang[2], Wilhelm Furnon[1], Nicolas M. Suarez[1], Daniel Mair[1], Ana da Silva Filipe[1], Brian J. Willett[1], Sam J. Wilson[1], Arvind H. Patel[1], Emma C. Thomson[1], Massimo Palmarini[1], Alain Kohl[1], Meredith E. Stewart[1☯*]

1 MRC-University of Glasgow Centre for Virus Research, Glasgow, United Kingdom, 2 Institute of Technology, University of Tartu, Tartu, Estonia

☯ These authors contributed equally to this work.
¤ Current address: Institut für Medizinische Virologie, Zurich, Switzerland
* Meredith.Stewart@glasgow.ac.uk

**Data Availability Statement:** SARS-CoV-2Engl2 was supplied under a MTA between The University of Glasgow and Public Health England. Consensus sequences and raw FASTQ files have been

## Abstract

Remdesivir (RDV), a broadly acting nucleoside analogue, is the only FDA approved small molecule antiviral for the treatment of COVID-19 patients. To date, there are no reports identifying SARS-CoV-2 RDV resistance in patients, animal models or *in vitro*. Here, we selected drug-resistant viral populations by serially passaging SARS-CoV-2 *in vitro* in the presence of RDV. Using high throughput sequencing, we identified a single mutation in RNA-dependent RNA polymerase (NSP12) at a residue conserved among all coronaviruses in two independently evolved populations displaying decreased RDV sensitivity. Introduction of the NSP12 E802D mutation into our SARS-CoV-2 reverse genetics backbone confirmed its role in decreasing RDV sensitivity *in vitro*. Substitution of E802 did not affect viral replication or activity of an alternate nucleoside analogue (EIDD2801) but did affect virus fitness in a competition assay. Analysis of the globally circulating SARS-CoV-2 variants (>800,000 sequences) showed no evidence of widespread transmission of RDV-resistant mutants. Surprisingly, we observed an excess of substitutions in spike at corresponding sites identified in the emerging SARS-CoV-2 variants of concern (i.e., H69, E484, N501, H655) indicating that they can arise *in vitro* in the absence of immune selection. The identification and characterisation of a drug resistant signature within the SARS-CoV-2 genome has implications for clinical management and virus surveillance.

## Author summary

The emergence of SARS-CoV-2 has led to a worldwide pandemic with significant morbidity and mortality. Remdesivir is the only antiviral with FDA approval for treatment. Antivirals use comes at a risk, as viruses may acquire mutations to overcome the inhibition. We identified a mutation in the virus polymerase responsible for decreased sensitivity to

uploaded to GenBank under BioProject number PRJNA692078. We used the publicly available CoV-Glue database (http://cov-glue.cvr.gla.ac.uk/#/home) to examine for replacements at specific sites observed in the GISAID hCoV-19 sequences. All relevant data within the manuscript and its supporting information are available on Enlighten: Research Data (doi.org/10.5525/gla.researchdata.1184).

**Funding:** This work was supported by the UK Medical Research Council (MC_UU_12014/8, MC_UU12014/2 and MC_UU_12014/12). RJO was funded by MC_UU_12014/12, AM & SW were funded by European Regional Development Fund, Centre of Excellence in Molecular Cell Engineering, Estonia (2014-2020.4.01.15-013), AMS was funded by UKRI/DHSC (BB/R019843/1) and MES by European Commission Horizon2020 (H2020-EU.3.2.1.1). The funders had no role in study design, data collection and analysis, decision to publish, or preparation of the manuscript.

**Competing interests:** The authors have declared that no competing interests exist.

Remdesivir. A change at this conserved site was not predicted, and the mutation did not cause a replication advantage or change in sensitivity to another antiviral drug. Importantly, this change occurred at very low frequency globally. Unexpectedly, passage of SARS-CoV-2 led to an accumulation of mutations in spike. A number occurred at the same sites but to different residues as those in emerging variants of concern indicating they arise in the absence of immune pressure. Our data indicates low-level Remdesivir resistance in SARS-CoV-2 is different to other RNA viruses and monitoring changes *in vitro* provides insight into general virus adaptation of newly emerging viruses.

## Introduction

The COVID-19 pandemic has caused more than 4 million deaths by July, 2021 [1] and placed the global economy under considerable strain [2]. The global effort to repurpose antiviral inhibitors and anti-inflammatory compounds to stem virus replication and clinical pathology identified Remdesivir (RDV or Veklury), a broadly acting nucleoside analogue, as a frontline treatment for patients hospitalized with severe acute respiratory syndrome virus-2 (SARS-CoV-2), the infectious agent underlying COVID-19. It is the only small molecular inhibitor that has FDA approval for the treatment of hospitalized SARS-CoV-2 patients. RDV exhibits a potent ability to restrict highly pathogenic human coronaviruses including severe acute respiratory syndrome virus (SARS-CoV), Middle eastern respiratory virus (MERS-CoV) as well as SARS-CoV-2 and Ebola virus (EBOV) replication *in vitro* [3,4] and in pre-clinical animal models [4–6]. Three randomized trials [7–9] demonstrated that RDV treatment reduced recovery time by 31% and demonstrated a non-significant trend towards lower mortality, thus reducing long-term healthcare costs. This trend of reduced hospitalization time and decreased morbidity was further supported by smaller non-randomized studies [10]. Conversely, a larger trial conducted by WHO (Solidarity Therapeutics Trial) reported no effect on patient survival [11]. The timing of administration of RDV appeared to be critical for its efficacy [4–6]. Despite these findings, countries including the USA and UK routinely use RDV for the treatment of hospitalized SARS-CoV-2 patients requiring oxygen who are still within the virological phase of infection (<10 days of illness). RDV is often prescribed in combination with dexamethasone, a steroid treatment, which reduces mortality in ventilated patients [12,13]. However, RDV and dexamethasone have yet to be trialed in combination. Recently RDV use in combination with Baricitinib has shown great promise [14].

Viruses often adapt and mutate to become resistant to antiviral therapy and this can affect patient and disease management. This is exemplified by viruses including human immunodeficiency virus type 1, hepatitis C virus, and influenza A which have all shown the ability to develop resistance during single drug use therapies [15–18]. Currently, there are no reports of circulating RDV-resistant strains of SARS-CoV-2. We are reliant on models based on studies in murine hepatitis virus (MHV), severe acute respiratory syndrome virus (SARS-CoV) and Ebola virus (EBOV) [19–21] in order to predict the changes in amino acid residues that could confer drug resistance. Given the global threat presented by SARS-CoV-2, it is important to determine whether SARS-CoV-2 can become resistant to RDV, identify which mutations confer resistance, monitor the emergence of such variants in the population and adapt treatments in COVID-19 patients.

## Results

### Change in SARS-CoV-2 RDV susceptibility after serial passage

After determining optimal culture conditions (S1 Fig), SARS-CoV-2$_{Engl2}$ was passaged serially in either 1μM or 2.5μM RDV-supplemented media for 13 passages (SARS-CoV-2$_{Engl2}$ was isolated in February 2020; Fig 1A). Viruses serving as controls were passaged in parallel in either DMSO or media to monitor for mutations due to cell culture adaptation unrelated to RDV resistance. In total, we passaged SARS-CoV-2$_{Engl2}$ in parallel in 24 distinct cultures with different selective pressures (4 different conditions and 2 different virus inputs; Fig 1A). We monitored for cytopathic effect (CPE) during passaging of the cultures. CPE was observed in 7 of the 12 lineages passaged in RDV, with the loss of 5 lineages between p1 and p4 (Fig 1A and 1B). We observed general adaptation of the viruses to VeroE6 cells, with an increase in overall viral titers by 0.5 to 1 log$_{10}$ (Fig 1B) as well as a change in plaque phenotype (S2A Fig) after 13 passages.

Next, the replication kinetics and change in the half maximal effective concentration (EC$_{50}$) of RDV in a subset of passaged virus populations (Rem2.5p13.5, DMSOp13.5 and Mediap13.4) were assessed. Rem2.5p13.5 alone actively replicated in the presence of 7.5μM RDV (Fig 1C). However, titers in the presence of RDV were lower than those grown in the absence of RDV. Titers of control viruses, DMSOp13.5 and Mediap13.4 were consistently 5 log$_{10}$ lower when grown in the presence of RDV (Fig 1C). The Rem2.5p13.5, DMSOp13.5 and Mediap13.4 lineages displayed similar replication kinetics when cultured in the absence of RDV (Fig 1C). Furthermore, Rem2.5p13.5 lineage required a greater concentration of RDV to protect infected cells (S2B Fig) corresponding to a 2- to 2.5-fold increase in RDV EC$_{50}$ over a range of virus inputs in comparison with DMSOp13.5, and Mediap13.4 (S2C Fig). The partial resistance to a nucleoside analogue was specific for RDV, as we observed a minimal change in EC$_{50}$ of a second nucleoside analogue (EIDD2801), when comparing Rem2.5p13.5 (EC$_{50}$ ~9.1μM) to SARS-CoV-2$_{Engl2}$ (EC$_{50}$ ~8.9μM) (Figs 1D and S3C and S3D andS1 Text).

Subsequent analyses identified a second lineage, Rem1p13.5 with reduced sensitivity to RDV (Fig 1E). The EC$_{50}$ of Rem1p13.5 (~0.83μM) was comparable to Rem2.5p13.5 (~0.88μM), this was a 3.5- to 3.7-fold increase in EC$_{50}$ from the parental virus (EC$_{50}$~0.23μM). The RDV EC$_{50}$ for virus passaged in either media alone (EC$_{50}$~0.29–0.32μM) or DMSO (EC$_{50}$~0.12–0.22μM) corresponded with EC$_{50}$ for the parental stock virus (Fig 1F). The changes in RDV sensitivity paralleled those previously reported for MHV, SARS-CoV and EBOV resistant viruses [4,20].

### Common mutations in two virus populations with partial resistance to RDV

Direct comparison of the consensus sequences from a subset of viruses passaged in different conditions with original SARS-CoV-2$_{Engl2}$ revealed two fixed non-synonymous mutations in lineages with decreased RDV susceptibility of at least 2-fold change in EC$_{50}$ in two independently generated populations (Rem1p13.5 & Rem2.5p13.5). These mutations were not present in viruses either passaged in absence of RDV, or the input virus (SARS-CoV-2$_{Engl2}$) or SARS-CoV-2$_{Wu1}$ (MN908947) (S1 Data). The first mutation was identified as glutamine to aspartate at amino acid 802 (E802D) in the RNA-dependent RNA polymerase (RdRp) NSP12 (Fig 2A). A glutamate at this position is highly conserved between all betacoronaviruses including SARS-CoV, MERS-CoV and unclassified sarbecoviruses (Fig 2B and S2A Data). The E802 mutation occurs within the palm sub-domains (T680 to Q815; Fig 2A), and in proximity to amino acids predicted to interact with newly synthesized RNA (C813, S814 and Q815 [22];

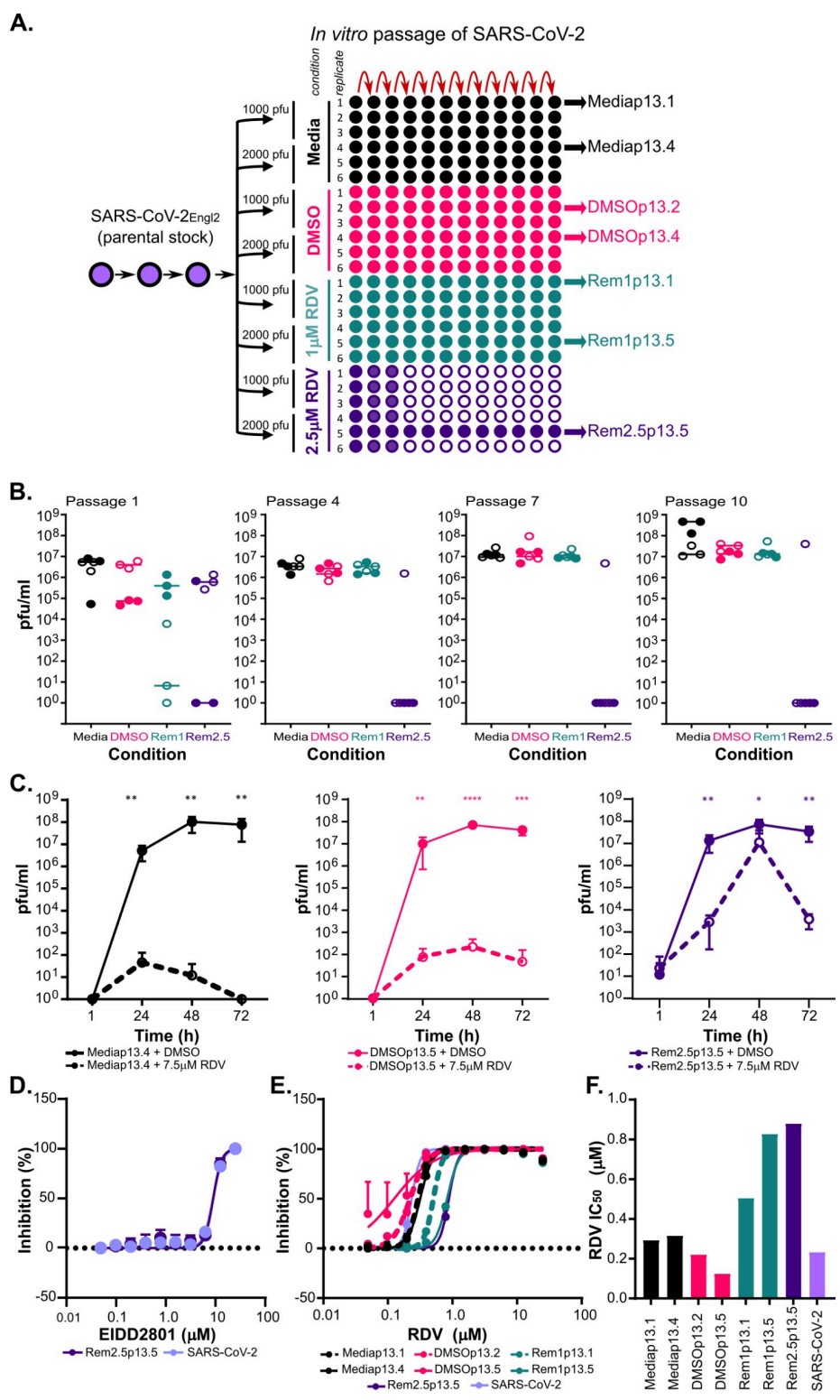

**Fig 1. Continuous passage of SARS-CoV-2$_{EngI2}$ in RDV selects for partial resistant populations.** **(A)** Schematic of the experimental layout to select for RDV resistant viruses. The passage (p) number of the input virus SARS-CoV-2$_{EngI2}$ is given. We consider p1 as the stock supplied by Public Health England, it was passaged twice in VeroE6 prior to selection in RDV. SARS-CoV-2$_{EngI2}$ p3 is the input stock that was sequenced and used for analysis. Each condition is given a different colour and the amount of virus used at the start of the experiment is indicated. Populations that failed to amplify (no CPE or/and virus titre) are indicated by the empty circles. All lineages sequenced are indicated. **(B)**

Virus titers (pfu/ml) at p1, p4, p7 and p10. 6 lineages per condition and two different virus inputs; 1000 pfu (solid circle) and 2000 pfu (open circle). Median for each is shown. **(C)** Virus growth kinetics in VeroE6 in the presence (dashed line) or absence (solid line) of 7.5µM RDV for 3 different virus populations. Data is from 2 independent experiments with 3 replicates. Error bars represent SEM. Unpaired t-tests (Holm-Šídák method; *,$P < 0.05$; **,$P < 0.01$; ***,$P < 0.001$. ****, $P < 0.0001$). **(D)** EIDD2801 dose dependency curve. EIDD2801 treated VeroE6-ACE2-TMPRSS2 infected with 8400 pfu/ml of each virus. **(E)** RDV dose dependency curves determined in A549NPro-ACE2 infected with 8400 pfu/ml of each virus. **(F)** Bar graph of RDV $EC_{50}$ for different viruses in A549NPro-ACE2 with 8400 pfu/ well. For all panels, error bars represent SEM.

Fig 2A). We propose that the E802D mutation changes the steric interactions between amino acid side chains within this region resulting in minor structural changes, (Fig 2A), thereby influencing binding of nt+3 during synthesis of template RNA and allowing elongation when the active form of RDV is incorporated into the RNA. The mutation identified in NSP12

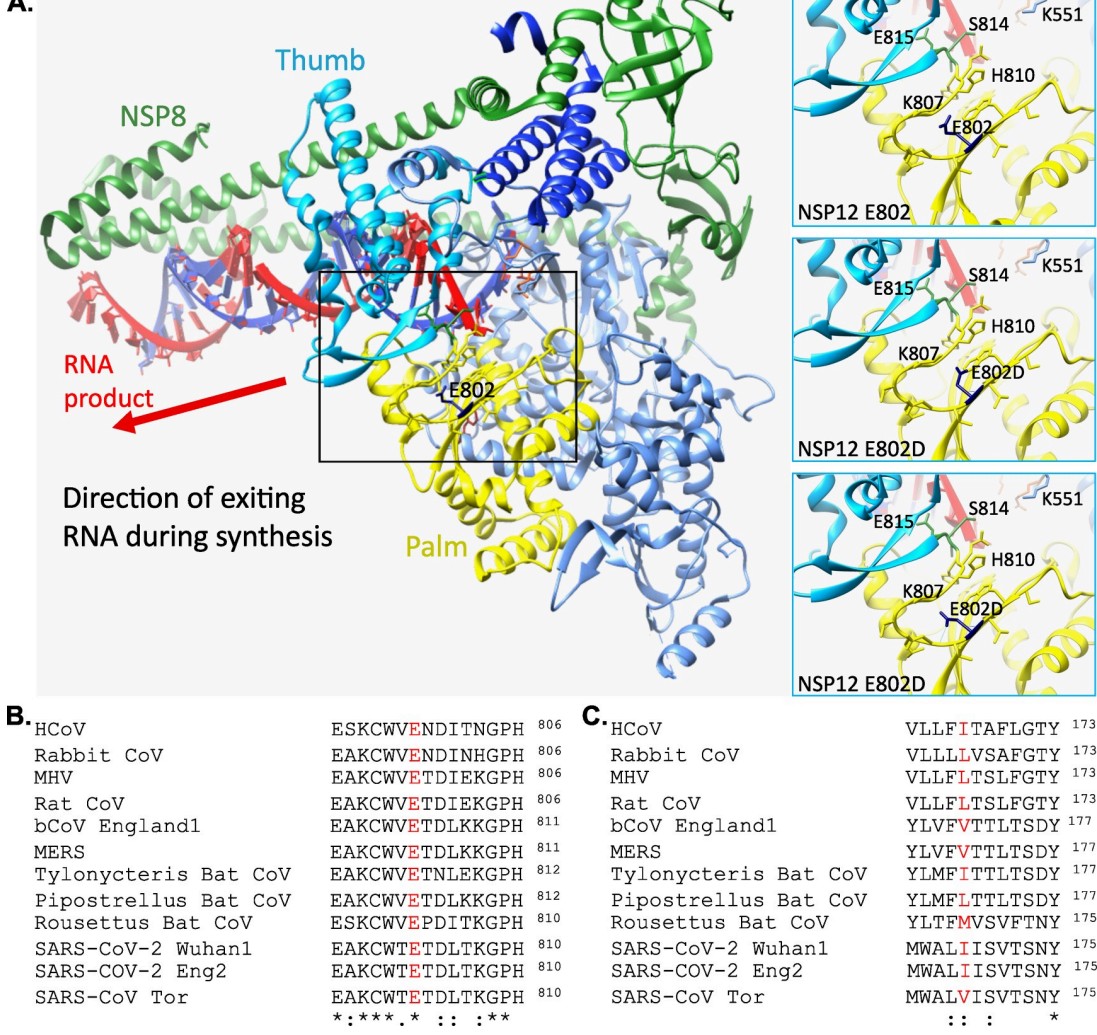

**Fig 2. Common mutations in partial RDV resistance populations. (A)** Location of E802 within structure of SARS-CoV-2 NSP12 in association with NSP7 and NSP8 (PDB ID 6YYT). Three focused panels are WT (upper) and two potential confirmations of E802D. H-bonds are indicated by light blue line. **(B)** Conservation of E802 amino acid across coronaviruses. Accession numbers for the coronavirus sequences are in the materials and methods. **(C)** NSP6 I168 amino acid is not conserved across coronaviruses.

differs from amino acid residues involved with decreased RDV sensitivity in EBOV, other betacoronaviruses, (MHV, & SARS-CoV), and sites predicted in SARS-CoV-2 [19–21].

The second mutation was an isoleucine to threonine substitution (I168T) in NSP6, a highly conserved protein involved in restricting autophagosome expansion [23]. This site is not conserved across coronaviruses with either an isoleucine (SARS-CoV-2), or valine (SARS-CoV & MERS-CoV) or leucine (MHV) in this position (Fig 2C and S2 Data). We predict this mutation may alter the structure of the transmembrane and extracellular domains (S4 Fig).

## A single mutation in NSP12 confer partial resistance to RDV in rSARS-CoV-2$_{Wu1}$

To ascertain whether a mutation of NSP12 E802 was sufficient to mediate partial RDV resistance, we introduced either an E802D or E802A mutation at this site into SARS-CoV-2$_{Wu1}$ by reverse genetics and recovered infectious virus. Whilst we thought it was unlikely to play a role, we also recovered virus with I168T mutation in NSP6 either alone or in combination with the NSP12 mutations (E802D or E802A). All rescued viruses with specific point mutations replicated similarly to the rescued wild-type virus (rSARS-CoV-2) in human lung cells, Calu-3, with comparable replication kinetics and achieving similar peak virus titers (Fig 3A; rSARS-CoV-2 vs rNSP12-E802D, p.adj = 0.44; rSARS-CoV-2 vs rNSP12-E802A, p.adj = 0.41; rSARS-CoV-2 vs rNSP6-I168T, p.adj = 0.51; rSARS-CoV-2 vs rNSP6-I168T+NSP12-E802A, p.adj = 0.50, rSARS-CoV-2 vs rNSP6-I168T+NSP12-E802D, p.adj = 0.50; RM One-Way Anova with Dunnett's Test). This indicates that these mutations did not provide a replicative advantage in a representative cell-line without drug exposure. In our drug screen assays, both the E802D and E802A mutations in NSP12 recapitulated partial resistance observed in the virus populations continually passaged in RDV (Fig 3B). We observed a 2.14- to 2.54-fold change in RDV EC$_{50}$; from 2.61μM for rSARS-CoV-2 to 5.58μM and 6.62μM for the E802A and E802D mutants, respectively (Fig 3B and S1 Table). This change in RDV sensitivity was evident over a range of virus inputs for both NSP12 mutants (S5A Fig). NSP6 I168T substitution did not confer decreased sensitivity to RDV (Fig 3B), with comparable EC$_{50}$ values to rSARS-CoV-2 (S1 Table).

Indeed, viruses bearing both the NSP12 and NSP6 mutations were more sensitive to RDV in comparison to NSP12 single mutant viruses (rNSP6-I168T+NSP12-E802D, EC$_{50}$ 3.21μM; rNSP6-I168T+NSP12-E802A, EC$_{50}$ 3.89μM). Importantly, introduction of NSP6 and/or NSP12 mutations did not significantly affect sensitivity to an alternate nucleoside analogue EIDD2801 (S1 Table and S5B Fig). These data confirm results obtained with other viruses demonstrating EIDD2801 sensitivity was not influenced by mutations conferring decreased RDV sensitivity [20,24]. We further assessed the anti-viral activity of RDV in Calu-3. While a RDV dose-dependent reduction in titer for all viruses was observed, rNSP12-E802D and rNSP12-E802A titers were consistently higher than wild-type and rNSP6-I168T at 24h and 48h pi (Fig 3C) over a range of concentrations. Interestingly, at 24h pi, a slight shift in an increase rNSP6-I168T infectious titer was observed in comparison with wild type, though this effect disappeared by 48h.

To examine whether the viruses containing the NSP12 E802D mutation would persist and outcompete the WT in the absence of RDV, an *in vitro* co-infection competition assay over multiple passages was undertaken. VeroE6-ACE2-TMPRSS2 were infected with two different MOI ratios of rSARS-CoV-2 to rNSP12-E802D (1:9 and 9:1), samples were harvested every 24 h and used to infect fresh monolayers of cells. This was repeated for 3 passages. While both stocks had high infectious titres (rSARS-CoV-2, ~2.5 x 10$^7$ pfu/ml and rNSP12-E802D, ~8.13 x 10$^7$ pfu/ml), there was a difference in the equivalent genome copies (GE)/ml, with

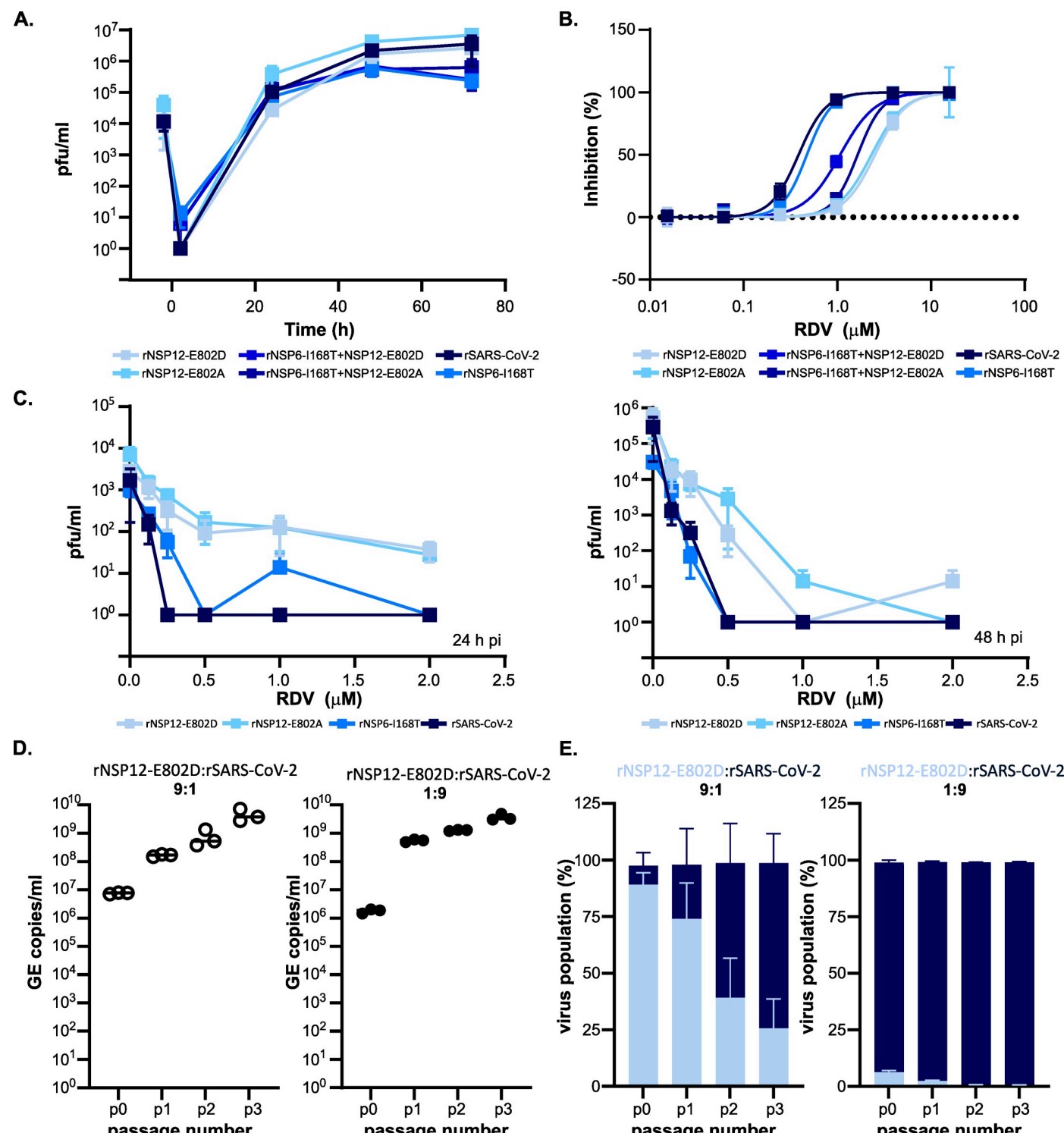

**Fig 3. NSP12 E802 mutation recapitulates change in RDV susceptibility.** All viruses were derived by reverse genetics and have a SARS-CoV-2$_{Wu1}$ backbone with specific point mutations as indicated. (**A**) Virus replication kinetics of rescued viruses with single mutation in either NSP12, NSP6 or both in Calu-3. Data is from 3 independent experiments with 3 replicates, there was no significant difference between growth of the mutants versus the wild-type rSARS-CoV-2. (**B**) RDV dose-dependent inhibition for each mutant virus. Post-treatment with RDV, VeroE6-ACE2-TMPRSS2 were infected with 500 pfu/ml of each virus. Mutations in NSP12 decrease the sensitivity to RDV. (**C**) RDV dose effect on virus titers at 24 h (left) and 48 h (right). Calu-3 were treated with decreasing doses of RDV and infected with an MOI~0.01. Data from 2 independent virus stocks with 2 replicates except for rSARS-CoV-2 and rNSP12-E802A. All error bars are SEM. (**D**) Co-infection competition assay virus titres for the input (p0) and each subsequent passage (p1, p2 and p3) expressed as genome copies/ml. VeroE6-ACE2-TMPRSS2 were

infected with different ratios of rNSP12-E802D to rSARS-CoV-2. The assay does not discriminate between rSARS-CoV-2 and rNSP12-E802D. Data from 3 independent infections. **(E)** Co-infection competition assay with different input ratios (1:9 and 9:1) of rSARS-CoV-2 WT and rNSP12E802D in VeroE6-ACE2-TMPRSS2. The percentage of each virus population over successive passages is shown (Data from 3 independent infections).

rNSP12-E802D a ~0.5 $\log_{10}$ higher (rSARS-CoV-2, 7.16 x $10^8$ GE/ml and rNSP12-E802D, 2.90 x $10^9$ GE/ml). Therefore, we used different ratios of MOI rather than GE/ml equivalents for the initial infection. We monitored for virus replication by qRT-PCR and found an increase in titres at each passage (Fig 3D). We next examined the change in the ratio of wild type to E802D in the viral populations at each passage using the GridION (Oxford Nanopore Technologies). Regardless of input ratio, rSARS-CoV-2 out competed viruses containing the rNSP12-E802D mutation (Fig 3E). This would indicate the E802D mutation incurs a fitness cost in the absence of RDV and is unlikely to be sustained in the circulating population.

## Prevalence of RDV mutations in circulating populations

We next examined the available SARS-CoV-2 genome sequences in the GISAID database (n = 844376 as of March 2021) to determine the frequency of replacements at NSP12 E802 and NSP6 I168. We only identified 36 (0.004%) viral sequences with a mutation at E802; E802D was the predominant replacement (n = 24), with the remaining sequence either E802A (n = 4), E802G (n = 3), E802Q (n = 1) or E802X (n = 1). The sequences containing these mutations were mostly geographically and temporally dispersed suggesting they did not share homology, though there was a small cluster of sequences (E802D, n = 9) from Germany within the same time frame. The 4 sequences with E802A substitution from May 2020 from the same geographic region (United Arab Emirates) but there is no evidence of further transmission. As one of these sequences, hCoV-19/Scotland/CVR2716/2020 was isolated from a patient who was not treated with RDV, this suggests mutation of E802 can arise in the community in the absence of drug selection.

The same pattern of very low observed global frequency of the E802 substitutions was also found at sites known to confer partial RDV resistance in other coronaviruses [20]. There were a handful of sequences with changes at either F480 (n = 13) or V557 (n = 35), and as with the E802 mutations, the sequences were mainly geographically and temporally dispersed. There was a cluster of sequences containing the V557I substitution observed in the Netherlands within a week but there was no information on whether it was the same patient or whether they were treated with RDV. When examining these mutations in respect to the global amino acid diversity along in NSP12 sequence, the region around E802 shows greater diversity than the area around F480 or V557 (S6A Fig). This suggests that strong purifying selection is acting in the regions of these sites, and they maybe more constrained by other dimensions of selection *in vivo*.

Replacement of NSP6 I168 occurred in 347 sequences with isoleucine replaced with threonine, valine, leucine, glycine or methionine. These data indicate that in absence of widespread selective pressure, substitutions of either NSP12 E802 or NSP6 I168 are currently rare global events. However, the identification of these sequences in the genome databases demonstrate that these viruses are viable and could potentially acquire a resistant phenotype when a stronger selective pressure is applied.

## *In vitro* mutations provide insight into spike adaption

We next focused on those mutations arising in the *in vitro* passaged virus populations that were likely not directly linked to RDV resistance. The consensus sequences of a subset of the

passaged stocks displayed a total of 41 distinct non-synonymous mutations and 10 synonymous mutations across the genome compared to the parental SARS-CoV-2$_{Engl2}$ sequence (Fig 4A). Importantly, we did not observe any previously identified mutations in the proof-reading ExoN (NSP14) that would change the sensitivity of the virus to RDV (Fig 4A and S1 Data). Deletions of ExoN have been demonstrated to increase RDV sensitivity for other coronaviruses [20]. While there was clear positive selection pressure across the entire genome (S2 Table), there were no major differences in the number of mutations that accumulated in any specific population, and in the ratio or type of transition vs transversion change (Figs 4B and S6B). Although, Rem2.5p13.5 displayed a slight elevation in non-synonymous changes (Fig 4C), we are unable to draw conclusions on the effect of RDV concentration on virus mutation rate due to small number recovered populations selected in RDV (S1 Text). Further comparative analysis over with the SARS-CoV-2$_{Wu1}$ sequence identified a further 2 synonymous and 8 non-synonymous mutations present in the original SARS-CoV-2$_{Engl2}$ population (Fig 4A). SARS-CoV-2$_{Engl2}$ was a 50:50 mix of two virus populations with 5 of the mutations present at a frequency ~50%, all but one of these became fixed in all passaged populations by p13 (Fig 4A).

Most of the collective 22 mutations observed in all sequences occurred within the spike (S) open reading frame. All of the *in vitro* non-synonymous substitutions appeared within multiple lineages the circulating viruses (GISAID data accessed March 2021, S3 Data) with the exception of I68R, N709H and D985G. Unlike observations in other studies [25–27], the furin-like cleavage site was preserved in all but one of the passaged populations. DMSOp13.2 displayed a 24nt deletion of the entire furin-like cleavage site at a high but unfixed frequency of 77% after 13 passages (S1 Data). Importantly, in our *in vitro* passaged viruses we observed both synonymous and non-synonymous substitutions occurring at the same sites within spike (H69R, E484D, N501T, H655Y, P681P) as those identified in the emerging SARS-CoV-2 variants of concern (Alpha (B.1.1.7): Δ69/70, N501Y, P681H; Gamma (P.1): E484K, N501Y, H655Y; Beta (B.1.351): E484K, N501Y) (Fig 4). Except for synonymous P681P, these substitutions were not present in SARS-CoV-2$_{Engl2}$ (S1 Data and S1 Text).

It is important to stress these emerging variants of concern, collectively, share a combination of three amino acid mutations in spike receptor binding domain (RBD): N501Y common to all and K417N and E484K in the Brazil and South African variants. We observe substitutions at 2 out of 3 of these sites in our *in vitro* evolution lines. The probability of such a large overlap (5 codons) between the substitutions observed *in vitro*, and variants of concern defining mutations without a common selective pressure driving convergence, was exceptionally small (P = 3.1x10$^{-5}$; S6C Fig). This demonstrates commonality in the fitness landscape that these *in vitro* populations and the circulating lineages are evolving under (driving evolutionary convergence). We further examined the global distribution of all circulating amino acid replacements within spike to determine whether our *in vitro* substitutions occurred within hot spots for change. There were 1384 replacements observed in a minimum of 5 sequences (n = 242865 sequence up dated 14$^{th}$ December 2020), many of these were clustered into certain regions within spike, creating visible hot spots of diversity (Fig 4E). For example, the window surrounding amino acid E484 appears to be a relative hot spot for replacement.

Of note are the four spike substitutions which occurred within the receptor binding domain (RBD) at either consensus (E484D, & N501T) or sub-consensus (G413R & Q498H) frequency. The E484 substitution appeared in the consensus sequence of Rem2.5p13.5 and Remp1p13.1 (Fig 4A), but it was present at a frequency of 20–40% in all the other viruses apart from DMSOp13.2 (S1 Data). The N501 substitution was present in one virus at consensus (Mediap13.1) and the sub-consensus of a second (Rem2.5p13.5) (Fig 4D). We noted that some *in vitro* substitutions in spike were changes to amino acids associated with SARS-CoV-1 spike

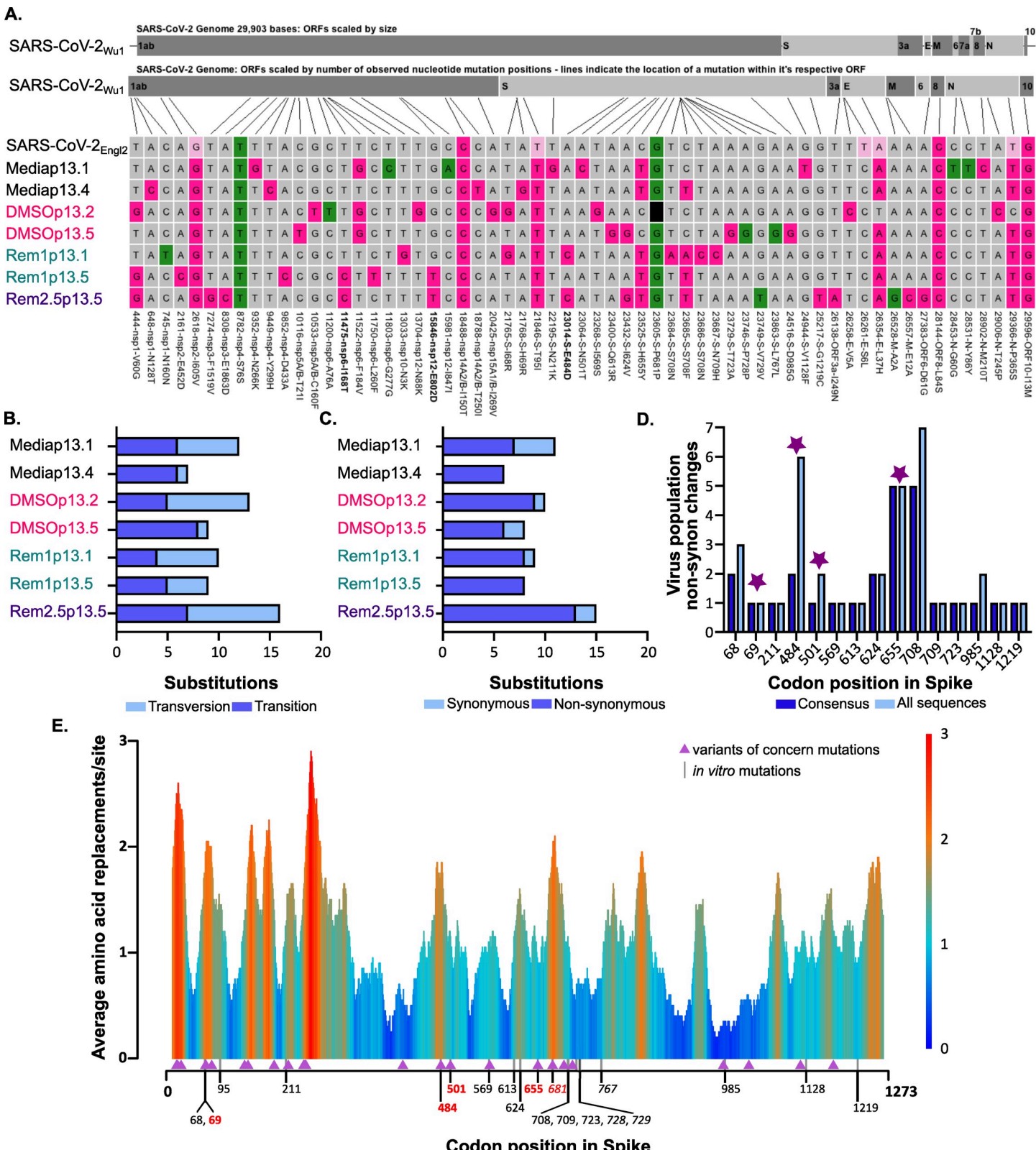

**Fig 4. Sequence analysis of continuous *in vitro* passaged SARS-CoV-2_Engl2.** **(A)** Alignment of serially passaged viruses and SARS-CoV-2_Engl2 to SARS-CoV-2_Wu1. Non-synonymous (pink) and synonymous (green) changes from Wuhan-1 are highlighted. Light pink are sites fixed at 50% in SARS-CoV-2_Engl2 and black box is a deletion mutation in serially passaged virus. Positions of mutations are indicated, and mutations only found in RDV selected populations are in bold. **(B)** Synonymous vs non-

synonymous changes observed in continually passaged virus populations compare to input SARS-CoV-2_Engl2. **(C)** Transversion vs transitional changes observed in continually passaged virus populations compare to input SARS-CoV-2_Engl2. **(D)** Number of *in vitro* passaged viruses with non-synonymous changes in Spike in comparison to SARS-CoV-2_Engl2. Mutation fixed in the consensus genomes (dark blue) are compared to the total number of viruses with evidence of the mutation at sub-consensus levels (light blue). Amino acid residues in common with the emerging variants of concern (Alpha (UK, B.1.1.7), Gamma (Brazil, P.1); and Beta (South Africa, B.1.351) are highlighted by a star. **(E)** Worldwide diversity of Spike protein sites of circulating SARS-CoV-2 variants. The average number of different substitutions at each codon is calculated along a 20 amino acid residues wide sliding windows. Data was calculated using only substitutions observed in a minimum of 5 sequences from the publicly available SARS-CoV-2 genomes (n = 1384). The position of the amino acid substitutions in the *in vitro* passaged viruses are indicated at the bottom, residues are red are shared with variants of concerns, black are specific to *in vitro* virus, residues in italics were synonymous. Mutations in amino acid residues that are also mutated in the variants of concern Alpha (UK, B.1.1.7), Gamma (Brazil, P.1), & Beta (South Africa, B.1.351) are shown in purple triangles.

(Q498H & N501T) adaption. Our data shows these mutations arise *in vitro* in the absence of any immune selection suggesting intrinsic fitness increases unrelated to immune evasion.

## Discussion

To our knowledge there are no reports identifying signatures within the genome of SARS-CoV-2 which lead to full, or even partial resistance to RDV. The change in sensitivity to RDV we observed was similar to single NSP12 mutation in either MHV or SARS-CoV [20] but lower than observed for the EBOV RdRp mutation [21]. It is of note that our partially-resistant RDV populations arose rapidly, with at least one lineage appearing within 4 passages (Rem2.5 lineage) and a second by passage 13 rather than 30 passages as observed for MHV, respectively [20,21]. However, RDV resistance mutations appeared in the EBOV genome by passage 9 but did not fix until passage 23 [21].

The potential for resistance to rapidly occur in RDV non-responding patients may be an issue that needs to be examined in order to discern whether it is due to a genomic mutation or drug tolerance by synchronization [28]. In a clinical setting, there has not been any evidence of RDV resistance arising in patients. Two studies showed clearance of viraemia after the second treatment but not after the first [29,30]. In the study reported by Kemp et al., [31] SARS-CoV-2 virus was never cleared or suppressed by any of the multiple treatments used and they speculated that a V157 mutation may be responsible for the resistance. This mutation has not been observed in coronaviruses including SARS-CoV-2 passaged in RDV *in vitro* or *in vivo* [20,21,29,32]. These studies followed the 10-day treatment RDV regime for the clearance of SARS-CoV-2 and the development of resistance may require longer treatment periods in patients before this phenotype becomes noticeable.

The *in vitro* mutations associated with partial RDV resistance for SARS-CoV-2 as well as those identified in MHV and SARS-CoV-1, circulate at extremely low frequency. This could in part be attributed to the decrease in fitness of coronaviruses with these RdRp mutations, as observed in competition assays results [20,21]. Indeed, we found that SARS-CoV-2 NSP12 E802D were outcompeted by wild type virus in our *in vitro* assays. This is unlike RdRp mutations in EBOV that conferred partial resistance and where the frequency was maintained in competition assays [21].

The action of nucleoside analogues to restrict coronaviruses is often thwarted by the unique proofreading activity encoded by the virus [20,33,34]. It has been demonstrated RDV functions by stalling the incorporation into the extending RNA product due to a steric clash with amino acid residue S861 *in vitro* [3,35,36]. However, it is possible for template-arrest caused by the incorporation of the nucleoside analogue and extension of the genome containing multiple RDV residues to occur [3,37]. The interplay between balance of naturally occurring dinucleotide concentration [3,37] and the cell's ability to metabolize RDV to its active form [4] are likely to influence whether break-through replication will occur. The SARS-CoV-2 NSP12 E802D mutation, similar to the NSP12 F480 mutation in MHV and SARS-CoV-1 does not fall

within the active sites, and we propose that this amino acid substitution results in minor structural modification of NSP12 that potentially influences the position of C813 and S814. These amino acids residues have a direct interaction with the newly formed RNA product [22,38,39] and recently have been shown to be critical for maintaining structural integrity of the NTP entry tunnel and reverse motion of the enzyme complex [40].

The SARS-CoV$_{Engl2}$ RDV-resistant (Rem1p13.5 and Rem2.5p13.5) and the reverse-genetic derived SARS-CoV-2$_{Wu1}$ NSP12 mutants (rNSP12_E802D & rNSP12_E802A) increased the EC$_{50}$ by at least 2-fold regardless of the cell type used for the experiments. Thus, we are confident that the change in EC$_{50}$ was not due to cellular drug metabolism or differences in virus entry and replication between wild-type and RDV-resistant viruses. We also noted that the cell-culture adaptation in viruses passaged in the absence of RDV resulted in a shift in EC$_{50}$ in VeroE6 based assays in comparison with input SARS-CoV-2$_{Engl2}$ (S3A and S3B Fig) but this shift was not as predominant as the RDV-selected viruses. Difference in EC$_{50}$ due to adaptation, availability of receptors and ability to metabolize RDV is widely acknowledged [4,41]. We hypothesize that this was due to more efficient virus entry and spread as many of the mutations observed occurred within the spike protein.

Coronaviruses correct most errors that occur during genome replication. Therefore, if mutations arise and are fixed there must be a general benefit that could include inhibitor escape, fitness advantage [42–44], antigenic change to escape antibody restriction [45,46] and adaption to the host [47]. There was no evidence within our data that RDV acts as a mutagen driving spontaneous mutations nor is it associated with increased rate of genome mutation with MHV and Ebola [20,21]. Other nucleoside analogues including ribavirin and EIDD2801 have been shown to act as mutagens [48,49], whereas the template stalling action of the RDV limits the spontaneous mutations and therefore the changes we observed *in vitro* are an evolutionary fine tuning of the viral proteins to a new host.

In our study, we also observe Spike mutations which evolved *in vitro* occurred disproportionately in the same codon sites as the variants of concern defining mutations. Additionally, many were found in localized hotspots of global diversity. We accessed the deep mutation scanning published data by Starr et al., [43] to examine whether these changes would influence RBD and affinity to ACE-2. In this study, only N501T and Q498Y changes observed in our data were associated with an increased affinity for ACE-2, while the E484D substitutions has a negative effect on ACE-2 binding in this system. Whereas, in an analysis using the entire spike protein there was an increase the free energy of the RBD-ACE-2 complex with the E484D substitution, resulting in a more infectious virus [50].

There is strong evidence based on circulating strains and *in vitro* sequences that the RDB of spike is under strong positive selection pressure [51,52]. Our *in vitro* observations suggest the predictability of evolution of the SARS-CoV-2 genome and could provide greater insight into the evolutionary pathway the virus is undergoing to adapt to its new host. The high ratio of non-synonymous to synonymous mutations we observed in spike (S2 Table) was driven by positive selection and not neutral evolution. Recently published studies have also shown the emergence of mutations in SARS-CoV-2 spike during *in vitro* passage at the same residues we identified including E484 and H655 [53,54]. Our data suggests that the parallel emergence of some mutations in the Spike in the variants of concern were not necessarily due to immune-based selection pressure. This is further highlighted by two mutations that arise *in vitro*; N501T and H655Y, the first within the RDB and the latter outside of the RDB and toward the S1-S2 cleavage site. Mutation of N501 increased affinity to human ACE-2 [43,45] and is involved in host tropism (i.e., SARS-CoV-1 N501 equivalent T487 mutation increase spike binding from civet to human ACE-2 [55,56]). H655Y has been observed by others *in vitro* [57], associated with adaption to cats and hamsters [54,58] antibody escape [59] and, identified

in the sequence of super spreaders [60] as well as arising by convergent evolution in circulating lineages (eg., P1 and A.27 lineages). While the phenotypic importances of H655 mutation is unknown, Braun et al., [54] postulated it is unlikely H655Y improves spike fusion efficiency and entry. This may indeed be the case, as the H655 substitution was not present in the only passaged virus with the multi-basic cleavage site deletion (DMSOp13.2) present in our data set. While not in the scope of this paper, functional studies. related to change observed during *in vitro* evolution experiments can help to identify candidate amino acid residues likely to drive future viral adaptation are recommended.

Unlike other studies using VeroE6 [25,26], we did not observe the rapid loss of the multi-basic (furin cleavage) site in SARS-CoV-2 spike *in vitro*. The cleavage of spike into the S1/S2 subunit is critical for virus entry and can be processed by multiple proteases including furin, serine protease (TMPRSS2/4) and the cysteine proteases Cathepsin B/L [61–63]. The pathways used for entry by coronaviruses is dependent on either TMPRSS2 or Cathepsin L expression [63–65]. VeroE6 are generally TMPRSS2 deficient but over express and process Procathespsin L to the active cathepsin L in abundance [64]. We postulate that this over expression of Cathepsin L allows for the maintenance of the multi-basic cleavage site in our continually passaged SARS-CoV-2 lineages. Other studies have also shown that the proteolytic cleavage site is maintained *in vitro* in continual passage for SARS-CoV-2 [66–68].

In summary, we have identified in our *in vitro* evolution studies a genome signature in SARS-CoV-2 which leads to replicative advantage in the presence of RDV. In the US, RDV treatment is currently prescribed to at least half of all hospitalized SARS-CoV-2 patients (as of January 2021 [69]) and was reportedly widely used during the second wave in India. Our data demonstrates that selection of RDV resistance in SARS-CoV-2 can occur but there is no evidence of global spread of RDV-resistant strains as the RDV resistance mutation identified in our study provides a cost to viral fitness. In addition, we have also shown that key amino acid residues that have been identified in emerging variants of concerns in three different continents can occur *in vitro* in the absence of immune pressure. Overall, our study offers new perspectives for the surveillance of new SARS-CoV-2 variants and the clinical management of patients treated with RDV.

## Materials & methods

### Cell culture

African Green monkey kidney cells VeroE6 expressing ACE-2 (VeroE6-ACE2) alone and with TMPRSS2 (VeroE6-ACE2-TMPRSS2), BHK-21, and A549NPro expressing ACE-2 (A549NPro-ACE2) were grown in DMEM-Glutamax supplemented with 10% fetal calf serum (FCS; Gibco) and non-essential amino acids (NEAA; Gibco). VeroE6 were a kind gift from Professor M. Bouloy (Institut Pasteur, France), BHK were sourced from the ATCC (ATCC CCL-10) and, VeroE6-ACE2, and VeroE6-ACE2-TMPRSS2 have been described elsewhere [66] A549NPro cells (a kind gift from Prof. R. Randall, University of St. Andrews, UK) were transduced to stably express ACE-2 (A549NPro-ACE2) as described [66]. Calu-3, a human lung epithelial cell line was grown in DMEM Glutamax, supplemented with 20%(v/v) FCS and NEAA was sourced from the ATCC (ATCC HTB-55). All cells were maintained at 37°C in 5% (v/v) $CO_2$, humidified incubator.

### Viruses

All initial work with SARS-CoV-2 was undertaken with the clinical isolate SARS-CoV-2$_{Engl2}$ strain (hCoV-19/England/02/2020 (England-02), GISAID accession: EPI_ISL_407073) kindly supplied by Public Health England. Validation of the mutations was undertaken in rescued

SARS-CoV-2 based on Wuhan-Hu-1 (MN908947) sequence. All virus stocks were grown in VeroE6 cells as described, supplemented with 4%(v/v) FCS, and for the drug screens regardless of cell type the final concentration of FCS was 2%(v/v). For virus growth curves, FCS concentration was decreased to 2%(v/v) in VeroE6 while in Calu-3 FCS concentration was 10%(v/v).

## Plaque assays

SARS-CoV-2 titers were determined by plaque assay in either VeroE6 or VeroE6-ACE2-TMPRSS2. Cells were seeded at $2.5 \times 10^5$ cells/well in 12-well plates and plaque assays performed as described [66].The plates were fixed in 8% [w/v] formaldehyde in PBS, stained with Coomassie Blue staining solution (0.1%(w/v) Coomassie Brilliant Blue R-250; 45%(v/v) methanol; 10%(v/v) glacial acetic acid) and imaged with a photo scanner (Epson Expression 1680 Pro).

## Chemical inhibitors

RDV (GS-5734; HY-104077) and EIDD-2801 (HY-135853) were purchased from MedChemExpress, diluted to final concentration of 10 mM stock solutions in 100% (v/v) dimethyl sulfoxide (DMSO, Sigma) and stored at -80C.

## Plasmids, virus rescue and validation of RDV resistant viruses

Reverse genetics plasmids used to rescue SARS-CoV-2$_{Wu1}$ have been described previously [66]. Briefly, specific point mutations in SARS-CoV-2 cDNAs in fragment 3 (nt11475 T to C for NSP6 I168T) and fragment 5 (nt 15845 A to C and nt 15846 G to C for NSP12 either E802A, or nt15846 G to C for NSP12 E802D) were synthesized by Biobasic and assembled into the full-length molecular clone to give pCCI-4K-SARS-CoV-2-NSP12_E802D, pCCI-4K-SARS-CoV-2-NSP12_E802A, pCCI-4K-SARS-CoV-2-NSP6_I168T, pCCI-4K-SARS-CoV-2-NSP6_I168T-NSP12_E802D-NSP6, pCCI-4K-SARS-CoV-2-NSP6_I168T-NSP12_E802D-NSP6. The pCCI-4K based plasmids containing the full-length SARS-CoV-2$_{Wu1}$ sequence with the mutations were propagated in TransforMax™ EPI300™ Electrocompetent *E. coli* and purified by using a Macherey Nagel Endotoxin free plasmid purification kit. For virus rescue, BHK-21 cells were transfected with 3 ug of plasmids using Lipofectamine LTX reagent as per manufacturer's recommendation. Cell culture supernatant was harvested 48h and 72h post transfection (p0) and used to infected VeroE6 to generated p1 stock of the recovered viruses.

## SARS-CoV-2 compound sensitivity studies and EC$_{50}$ calculations

To test for a change in sensitivity to RDV of our continually passaged SARS-CoV-2$_{Engl2}$ viruses and reverse genetic derived viruses (rNSP12-E802D, rNSP12-E802A, rNSP6-I168T, rNSP6-I168T-NSP12-E802D, rNSP6-I168T-NSP12-E802A and rSARS-CoV), two different compound screen layouts were tested (S7 Fig). Cells were seeded at $1.2 \times 10^4$ cells/well for VeroE6, VeroE6-ACE2, and VeroE6-ACE2-TMPRSS2 and $1.3 \times 10^4$ cells/well for A549NPro-ACE2 cell lines in a 96-well format. Two-fold serial dilutions of RDV starting at either 25 or 50 μM were added to the cells and incubated for a minimum of 2h prior to addition of virus. Depending on plate layout, either a set amount of virus or 2-fold serial dilutions of the virus were added, and plates incubated in a 37°C humidified incubator in 5% $CO_2$ for 72h. For the mixed array plate layout, each drug dilution and virus combination were in triplicate on each plate and repeated twice. Plates were fixed in 8%(w/v) formaldehyde for 1h and stained in Coomassie Brilliant Blue solution. Washed and airdried plates were scanned using the Celigo

(Nexcelcom) and the total intensity of each well was determined. The total intensity values from the Celigo scan were used to calculate the $EC_{50}$ values using Prism GraphPad v8 and v9. Analysis programs used included transformation of concentration ($log_{10}$), normalization of transformed data and non-linear regression (log(agonist) vs. normalized response—variable slope).

## Selection of RDV resistant viruses

Continual passages of SARS-CoV-2 were performed in the presence of RDV in VeroE6 cells. Assays were undertaken in a 12 well plate with each condition in triplicate; two RDV concentrations (1μM and 2.5μM) and two virus inputs (1000 & 2000 pfu) were used, and controls for cell culture adaption of the virus including media alone or 0.2%(v/v) DMSO were undertaken in parallel (Fig 1). These concentrations were kept constant throughout the experiment. Time between each amplification was dependent of visible CPE from passage 1 to passage 5 (p1 to p5), there was up to 7 days between each amplification. For these passages, 100 μl of each amplification was used to generate the subsequent passage. It became evident after passage 4 (p4) that some of the conditions used resulted in loss of infectious virus. After passage 5, 10 μl of culture media was used for passaging the virus in wells that showed evident CPE. Where CPE was not evident, 100 μl of cell culture was blindly passaged. After each passage the cell media was harvested and stored at -80˚C. Samples from p4, p7, p10 and p12 were also harvested for titration and cell monolayer was harvested in TrizolLS for future analysis. Prior to sequencing, each condition was tested to determine its ability to grow in the presence of RDV.

## Virus replication kinetics

To compare the replication kinetics of the parental stock, cell culture adapted and RDV resistant mutant SARS-CoV-2, monolayers of VeroE6 cells were treated with 7.5μM RDV and infected at a MOI~0.1. The inoculum was removed after 1h, cells washed and fresh media containing either DMSO or RDV added. Growth curves in VeroE6 were undertaken twice in triplicate. rSARS-CoV-2$_{Wu1}$ and mutant viruses virus growth was also assessed in Calu3 cells. For this, Calu-3 were seeded 7 days prior to infection, media was changed every 2–3 days prior to infection. The cells were infected with a MOI~0.01 (based on VeroE6-ACE2-TMPRSS2 titres) for 1h, inoculum then aspirated, cells washed, and fresh media added. All infections for virus growth curves were asynchronous. Cell culture media from each well were collected at 24, 48 and 72h post infection (p.i.) and cell-free virus titers were determined by plaque assay in VeroE6 cells. Each experiment was performed at least thrice with a triplicate of the virus each time (mutant viruses n = 3, wt n = 6). Virus titers at each time were determined by plaque assay as described above.

To assess the influence of RDV on the growth of rSARS-CoV-2 and mutant viruses, Calu-3 were seeded 7 days prior to infection in a 24 well format (2 x 10$^5$ cells/well), media was changed every 2–3 days. Two hours prior to infection 2-fold serial dilutions of RDV from 2μM to 0.125μM were added to cells. Subsequently, cells were infected at a MOI of 0.01 asynchronously for 1h, inoculum aspirated, and media containing the appropriate amount if RDV added. Samples were taken 24h and 48h pi. Titers were determined by plaque assay. Data from 2 independent virus stocks with 2 replicates except we were missing one of the replicates for wild-type and a 48h pi replicated for rNSP12-E802A at 48h pi.

## RNA purification from virus stock

Clarified supernatant from p13 was added to TrizolLS and total RNA was extracted as described [70] for sequence analysis.

## Sequence of virus stock using Illumina MiSeq

Sequencing libraries composed of overlapping amplicons across the SARS-CoV-2 genome were prepared utilizing the ARTIC Network protocol (https://artic.network/ncov-2019). Amplicons were paired-end sequenced (2x250 nt) on an Illumina MiSeq as described previously [71]. Reads were trimmed with trim_galore (http://www.bioinformatics.babraham.ac.uk/projects/trim_galore/) and then mapped with Burrows–Wheeler Aligner [72] to the SARS-CoV-2$_{Wu1}$ (MN908947) genome, followed by amplicon primer trimming and consensus calling with iVar [73] using a minimum read coverage of 10. Consensus sequences and raw FASTQ files have been uploaded to GenBank under BioProject number PRJNA692078.

## Co-infection competition assay

VeroE6-ACE2-TMPRSS2 were co-infected with different ratios (1:9, and 9:1) of rNSP12E802D and rSARS-CoV-2 WT to assess virus fitness as described by Agostini et al., (2018) [20]. Briefly, a 1:9 ratio was equivalent of MOI ~ 0.01 to a MOI of ~0.09, cells were infected for 1 h at 37˚C, media aspirated, cell washed, and fresh media added. Twenty-four h post-infection, virus supernatant was harvested, and 100 μl (10% of the volume) was used to infected new VeroE6-ACE2-TMPRSS2 and the infection procedure repeated for 3 passages. RNA was extracted from the clarified supernatant after heat inactivation at 56˚C for 20 min using Beckman Coulter RNAdvance Blood kit and KingFisher automated extraction system as per manufacturer's protocols.

## Quantification of SARS-CoV-2 genome copies

To examine the virus growth kinetic during the competition assay, the viral load was determined by quantitative RT-PCR based on copies of N RNA transcript. All samples were quantified using a NEB Luna Universal Probe One-Step RT-qPCR Kit (New England Biolabs, E3006) and a previously described CDC 2019-Novel Coronavirus (2019-nCoV) Real-Time reverse transcription PCR Panel [74,75]. 2019-nCoV N1 forward primer 5'-GAC CCC AAA ATC AGC GAA AT-3', 2019-nCoV N1 reverse primer 5'-TCT GGT TAC TGC CAG TTG AAT CTG-3' and 2019-nCoV N1 probe 5'-FAM-ACC CCG CAT TAC GTT TGG TGG ACC-BHQ1-3' were part of the SARS-CoV-2 (2019-nCoV) CDC RUO Primers and Probes kit (IDT Technologies, 10006713). SARS-CoV-2 N RNA was quantified using a ten-fold serial dilutions of an in-house SARS-CoV-2 RNA standard to enable genome equivalent (GE) copies/ml as the is a high correlation between copy number of N and ORF1ab as targets [76] from SARS-CoV-2 RNA extracted from supernatant. All reactions were performed on ABI 7500 Fast thermal cycler.

## Sequence of virus stock using GridION (Oxford Nanopore Technology)

The GridION system was used to sequence the virus populations used for the competition assay as previously described [71]. GridION reads were processed using the ARTIC bioinformatics pipeline (https://github.com/artic-network/artic-ncov2019). Briefly, reads were size filtered then aligned to the Wuhan-Hu-1 reference (MN908947) with Minimap2 [77] followed by post-alignment amplicon primer trimming and variant calling with Nanopolish (https://github.com/jts/nanopolish).

## Modelling of mutations in SARS-CoV-2 NSP12

UCSF Chimera [78] to generate the ribbon structure and rotamer function [79] used to model the mutations into SARS-CoV-2$_{Wu1}$ NSP12 (PDB 6YYT [39]). Protter [80] was used visualize the effect of the I168T mutation on organization of the transmembrane domains in NSP6.

## Statistics and bioinformatic

GraphPad Prism v8 & v9 software (La Jolla, CA, USA) was used to undertake statistical analysis (transformation, normalization, non-linear regression, RM one-way ANOVA (Dunn Dunnett's multiple comparisons test), two-way ANOVA, unpaired t-test (Holm-Šídák method)) and generate figures. EMBL-EBI Clustal omega [81] was used to align the sequences of NSP6 and NSP12 from SARS-CoV-2 Wuhan1 (MN908947), betacoronavirus England1 (bCoV England1; YP009944302 & YP009944297.1), SARS-CoV-2$_{Engl2}$, SARS-CoV Tor (NP_828849.7 & YP009944371.1), human coronavirus (hCoV, HKU1, YP459941.1 & YP009944274.1), middle eastern respiratory virus (MERS; YP009047223.1 &009047234.1), murine hepatitis virus (MHV; YP009924352.1 & 009915678.1), rat coronavirus (Parker, YP009924378.1 & YP_009924378.1), Rousettus bat coronavirus (HKU9; YP009924393.1 & YP009924388.1), rabbit CoV (HKU14; YP009924419.1 & YP009924414.1), Tylonycteris bat CoV (HKU4 YP009944320.1 & YP009944315.1), and Pipostrellus bat CoV(HKU5; YP009944349.2 & YP009944344.1). The online database CoV-GLUE (http://cov-glue.cvr.gla.ac.uk/[82]) was used to monitor for replacement in the SARS-CoV-2 genome. The database was accessed on 10th of January 2021 (n = 242865) for generating the diversity plot and further on 15$^{th}$ of March 2021 n = 426623). The dN/dS ratios were calculated for each sample using all ORFs concatenated together, as well as the Spike and entire ORF1ab sequences individually (with respect to the SARS-CoV-2$_{Engl2}$ sample to monitor within passage evolution), using the program SNAP (https://www.hiv.lanl.gov/content/sequence/SNAP/SNAP.html)[83].

The code used to calculate the null distribution and the probability as well as the global distribution of the mutations within in Spike can be assessed at https://github.com/omaclean/CV19IV/.

The receptor binding domain of Spike is from amino acids 319–541: B1.351 and P1 had 3 mutations in the RBD (K417N/T, E484K and N501Y) while Alpha (B.1.1.7) only had one (N501Y). We used the following amino acids replacements and deletions in Spike for each lineage-based COG-UK defined changes for each lineage; Beta (B1.351; L18F, D80A, D215G, R246L, K417N, E484K, N501Y, A701V), Gamma (P.1; L18F, T20N, P26S, D138Y, R190S, K417T, E484K, N501Y H655Y, T1027I), Alpha (B.1.1.7; Δ69–70, D144, N501Y, A570D, P681H, T716I, S982A, D1118H). We further included the substitution observed in P1 (V1176F) and deletion in B1.351 (Δ242–245). There is phylogenetic uncertainty if V1176F was present in the ancestor preceding the formation of the Gamma (P.1) lineage, the deletion (Δ242–245; like other defining mutations) in Beta (B.1.351) is polymorphic. We erred on the side of inclusivity in the face of this uncertainty, making our analysis more conservative. All sequences available sequences for SARS-CoV-2 were assessed from CoV-Glu (n = 242865, last update 14$^{th}$ December 2020).

## Supporting information

**S1 Fig. RDV mixed array inhibitor screen with SARS-CoV-2$_{Engl2}$ in VeroE6 cells.** Two-fold serial dilutions of RDV from 50μM and either a two-fold (left) or three-fold (right) dilution of SARS-CoV-2. **(A)** Heatmap of total intensity of the stained plated generated by the Celigo. Lightest colors indicate clearance of the monolayer. **(B)** RDV dose dependency over a range of virus inputs. For each RDV concentration the survival (%) of the monolayer with different virus inputs per well is plotted. Values are from 1 replicate per condition.
(TIF)

**S2 Fig. Adaptation of SARS-CoV-2$_{Engl2}$ to VeroE6 and selection of decreased RDV sensitivity. (A)** Coomassie blue stained SARS-CoV-2 plaque assays in VeroE6. Change in plaque

morphology from passage 1 and passage 10. A subset of populations from the 4 different conditions were compared. **(B)** Celigo scans of Coomassie blue stained 96-well format mixed array drug for two control populations (Mediap13.4 and DMSOp13.5) and a RDV adapted population (Rem2.5p13.5). RDV concentration and virus input is indicated. **(C)** Bar graph of RDV $EC_{50}$ for an RDV adapted population and two control populations over a range of virus inputs.
(TIF)

**S3 Fig. Antiviral activity of RDV and EIDD2801 in Vero-ACE-2 against three continually passaged populations and the input SARS-CoV-2$_{Engl2}$.** The $EC_{50}$ of RDV adapted population REM2.5p13.5 was compared to the input virus SARS-Cov-2$_{Engl2}$ as well as DMSOp13.4 and Mediap13.4. **(A)** RDV dose dependency curve. **(B)** Bar graph of RDV $EC_{50}$ required to protect the monolayer. **(C)** EIDD2801 dose dependency curve. **(D)** Bar graph of EIDD2801 $EC_{50}$ required to protect the monolayer.
(TIF)

**S4 Fig. Protter prediction transmembrane organisation of NSP6 WT (left) and NSP6 I168T (right) mutation**. Position of I168 and I168T are indicated by an arrow. Extra-, intracellular and membrane are indicated, and putative glycosylation site is shown in green.
(TIF)

**S5 Fig. Antiviral activity of RDV and EIDD2801against NSP6 and NSP12 mutant viruses.** All viruses have a SARS-CoV-2$_{Wu1}$ backbone with specific point mutations as indicated. Assay undertaken in VeroE6-ACE2-TMPRSS2. **(A)** Bar graph of RDV $EC_{50}$ values for rNSP12E902A, rNSP12E802D and rSARS-CoV-2 over a range of virus inputs. **(B)** Bar graph of EIDD2801 $EC_{50}$ for each rescued virus. Error bars are SEM. Experiment from 2 independent virus stocks with 3 technical replicates.
(TIF)

**S6 Fig. Sequence analysis of continually RDV selected SARS-CoV-2$_{Engl2}$. (A)** Diversity occurring in SARS-CoV-2 NSP-12 in 20 amino acid sliding window. Calculated from 988 mutations observed in minimum of 5 sequences. Position of mutation associated with decrease RDV sensitivity in SARS-CoV-2 is highlighted in red and sites identified in murine hepatitis virus (MHV) and Ebola virus (EBOV) are indicated in black. **(B)** Heatmap of the distribution of transversion and transition depending on nucleotide. **(C)** Distribution of convergence. The null distribution of overlapping *in vitro* mutations hitting the same Spike codon positions as in the SARS-CoV-2 variants of concern. 21 *in vitro* mutations and 20 variant of concern mutations spread across 1271 codons. 100 million simulations were run.
(TIF)

**S7 Fig. Schematic layout of 96-well plate clearance assay for drug screen.** All layouts have controls for inhibitor toxicity, virus clearance and cell monolayer integrity (DMSO only). **(A)** Mixed array layout with range of inhibitor dilutions vs a range of virus dilutions. **(B)** Mixed plate layout inhibitor is diluted and a set amount of virus. Amount of virus added causes complete well clearance by 72 h pi. The mixed plate layout can also be modified to dilute inhibitor down the rows rather than across the columns.
(TIF)

**S1 Table. RDV and EIDD2801 $EC_{50}$ and fold-change in VeroE6-ACE2-TMPRSS2.** All viruses have a SARS-CoV-2$_{Wu1}$ backbone with specific point mutations as indicated. Average values and SEM from independently rescued virus stocks, except for rNSP12 E802A. Experiment undertaken at least twice in triplicate by independent operators apart for rNSP6I168T in

the presence of EIDD2801 (n = 1) and n = 3 for rSARS-CoV-2, rNSP12E802D, rNSP12E802A and rNSP6I168T.
(XLSX)

**S2 Table. dN/dS ratio for all SARS-CoV-2 open reading frames, ORF1ab and spike for the *in vitro* continually passaged SARS-CoV-2.**
(XLSX)

**S1 Data. Excel sheet summarizing mutations across different continually passaged virus populations in comparison to SARS-CoV-2 $_{Wuhan-1}$ (MN908947) and SARS-CoV-2$_{Engl2}$. (A)** Key for the virus names **(B)** All the mutations occurring in the consensus sequence in the continually passaged viruses and in the input virus SARS-CoV-2$_{Engl2}$ in comparison to SARS-CoV-$_{2Wu1}$. **(C)** The frequency of all mutations (consensus and subconsensus) that arose in continually passaged SARS-CoV-2$_{Engl2}$ and **(D)** a list of the deletions and insertions.
(XLSX)

**S2 Data. Conservation of NSP12 E802 and NSP6 I168 in coronaviruses.** Excel file with the frequency of NSP12 E802 **(A)** and frequency of NSP6 I168 **(B)**.
(XLSX)

**S3 Data. SARS-CoV-2 spike mutation arising *in vitro* are prevalence in the circulating sequences.** The mutation counts (total of 3588809) and lineage frequency are generated from >800 000 sequences available on GISAID. Frequency >0.5 is the number of lineages the mutation is >50% of the total sequences within each lineage, whereas frequency <0.5 is the number of lineages the mutation appears where the frequency of that mutation is less than 50%. Lineages that the mutations appear in are indicated.
(XLSX)

**S4 Data. GISAID Acknowledgements.** Authors and laboratories contributing to the genome data to GISAID database used to identify the NSP12 E802 mutation in SARS-CoV-2. All Submitters of data may be contacted directly via www.gisaid.org. Contributing authors and laboratories are sorted alphabetically. The accession number of each sequence is provided along with the originating laboratory.
(XLSX)

**S1 Text. Supporting information contains additional information on the SARS-CoV-2 adaptation in VeroE6, expands on lack of mutations in the presence on RDV, and the pressure to evolve with greater focus on the Spike.**
(DOCX)

## Acknowledgments

We would like to thank all the global researchers who shared their data on GISAID (https://www.gisaid.org; S4 Data), COVID-19 Genomics UK Consortium (COG-UK) and Kyriaki Nomikou, Jenna Nichols, Yasmin Parr, Lily Tong and Natasha Johnson from the CVR Genomics Team.

## Author Contributions

**Conceptualization:** Agnieszka M. Szemiel, Emma C. Thomson, Massimo Palmarini, Alain Kohl, Meredith E. Stewart.

**Data curation:** Agnieszka M. Szemiel, Richard J. Orton, Oscar A. MacLean, Meredith E. Stewart.

**Formal analysis:** Agnieszka M. Szemiel, Meredith E. Stewart.

**Funding acquisition:** Andres Merits, Ana da Silva Filipe, Brian J. Willett, Sam J. Wilson, Arvind H. Patel, Massimo Palmarini, Alain Kohl.

**Investigation:** Agnieszka M. Szemiel, Andres Merits, Richard J. Orton, Oscar A. MacLean, Gauthier Lieber, Matthew L. Turnbull, Sainan Wang, Daniel Mair, Meredith E. Stewart.

**Methodology:** Agnieszka M. Szemiel, Andres Merits, Oscar A. MacLean, Rute Maria Pinto, Wilhelm Furnon, Nicolas M. Suarez, Ana da Silva Filipe, Sam J. Wilson, Arvind H. Patel, Meredith E. Stewart.

**Project administration:** Agnieszka M. Szemiel, Alain Kohl, Meredith E. Stewart.

**Resources:** Arthur Wickenhagen, Ana da Silva Filipe, Sam J. Wilson, Arvind H. Patel, Emma C. Thomson, Massimo Palmarini, Alain Kohl.

**Validation:** Agnieszka M. Szemiel, Oscar A. MacLean, Meredith E. Stewart.

**Visualization:** Agnieszka M. Szemiel, Richard J. Orton, Oscar A. MacLean, Meredith E. Stewart.

**Writing – original draft:** Agnieszka M. Szemiel, Brian J. Willett, Massimo Palmarini, Alain Kohl, Meredith E. Stewart.

**Writing – review & editing:** Agnieszka M. Szemiel, Andres Merits, Richard J. Orton, Oscar A. MacLean, Rute Maria Pinto, Arthur Wickenhagen, Gauthier Lieber, Matthew L. Turnbull, Brian J. Willett, Emma C. Thomson, Massimo Palmarini, Alain Kohl, Meredith E. Stewart.

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
