## [Decision Letter · Decision Letter 0]

28 May 2021

Dear Dr Stewart,

Thank you very much for submitting your manuscript "In vitro selection of Remdesivir resistance suggests evolutionary predictability of SARS-CoV-2" for consideration at PLOS Pathogens. As with all papers reviewed by the journal, your manuscript was reviewed by members of the editorial board and by several independent reviewers. In light of the reviews (below this email), we would like to invite the resubmission of a significantly-revised version that takes into account the reviewers' comments.

We cannot make any decision about publication until we have seen the revised manuscript and your response to the reviewers' comments. Your revised manuscript is also likely to be sent to reviewers for further evaluation.

Sincerely,

Shin-Ru Shih

Section Editor

PLOS Pathogens

Shin-Ru Shih

Section Editor

PLOS Pathogens

Kasturi Haldar

Editor-in-Chief

PLOS Pathogens

orcid.org/0000-0001-5065-158X

Michael Malim

Editor-in-Chief

PLOS Pathogens

orcid.org/0000-0002-7699-2064

Reviewer's Responses to Questions

**Part I - Summary**

Reviewer #1: Remdesivir (RDV) is the only antiviral with FDA approval for treatment of COVID-19 patients. In this study, the authors investigated the resistance of SARS-CoV-2 to RDV under different concentrations through serial passage of the virus in Vero-E6 cell line. The authors identified two amino acid mutations with one located in NSP12 and the other in NSP6. Through reverse genetics, the authors confirmed the one located in NSP12, E802, is the major mutation contributing to the loss of sensitivity to RDV. Upon plenty of studies shown the mechanism and interaction between virus and nucleoside analog (RDV), several amino acids either direct interacting with RDV, such as residue 687/6222/557/480, or by delayed chain termination, such as residue 861, have been identified. The mutation identified in this study, E802D, could result in resistance to RDV due to steric clash as suggested by the authors. However, this probably won’t be the only sites with resistance and the development of reverse genetic system to determine the EC50 will be important to identify the resistant variants from the global viral variant surveillance system, which is the merit of this study.

Reviewer #2: Szemiel and colleagues determine, in vitro, that resistance to Remdesivir can evolve in SARS-CoV-2. They find a recurrent substitution that is able to confer resistance. Their passage experiments also identify sites throughout the genome that are tolerant of diversity. Experiments are expertly performed and well controlled.

**Part II – Major Issues: Key Experiments Required for Acceptance**

Reviewer #1: 1. Many studies have shown multiple spontaneous mutations on spike protein when serial passage of SARS-CoV-2 on Vero-E6 cell line are under selective pressure (Ex. Sasaki M, Uemura K, Sato A, Toba S, Sanaki T, Maenaka K, et al. (2021) SARS-CoV-2 variants with mutations at the S1/S2 cleavage site are generated in vitro during propagation in TMPRSS2-deficient cells. PLoS Pathog 17(1):e1009233. Or Journal of General Virology 2020;101:925–940) These studies suggested that virus variants with spike gene mutations generated smaller plaques as shown in this study due to the loss of the ability to utilize the entry pathway involving TMPRSS2 and alerted the researchers propagating the virus in tissue culture without TMPRSS2. Therefore, the mutations on spike protein gene generated through serial passage of SARS-CoV-2 on vero-E6 cell might be due to (1) spontaneous mutations irrelevant to RDV, or (2) concordant mutations due to RNA secondary structure upon RDV treatment, or (3) change in receptor usage upon RDV treatment since the lack of ACE2 receptor on vero-E6, which may allow the virus to use other method of entry. Further experimental confirmation or detail discussions is demanded.

2. Certain mutant virus arises under selection pressure with the cost of fitness lost. Such mutant virus lost the growth under competition with the wild type virus. The authors should investigate any fitness change by mixing the mutant virus with the parental virus with different ratio.

3. Figure S1 should illustrate what the difference is between mock and 0 pfu/well. And why the input virus (1024-32pfu/well) at left and (14-1pfu/well) at right showed the lightest color representing clearance of monolayer, compared to high input viruses such as 32768pfu/ml?

4. Line 109-110, could the authors explain why Rem1p13.5 was selected for EC50 testing instead of testing all lineages which Rem1p13 survived?

5. From Figure 4, it is not clear if the authors sequenced all the lineages of survived viruses as mentioned in Line 116 by the authors. Since multiple spontaneous mutations could occur when serial passage of SARS-CoV-2 on Vero-E6 cell line, sequencing all the lineages could assure the results illustrated in section “Common mutations in two virus populations with partial resistance to RDV”. It is important since natural resistant viruses can occur under just DMSO treatment as shown in Fig 1C. If the authors did sequence all the lineages of viruses, a consensus map like figure 4 showing all the sequenced viruses will be better than just providing datafileS1.

6. Line 144-147, need to specify what kind of statistics was used to get p value and p-adj? And what p-adj mean? Meanwhile, Line 406-408, need to specify which statistics was used in which experiments?

7. Figure 3C, mutant virus rNSP12-E802D showed decrease in titer from 10^2 at 24hr to 10^1 at 48hr under 1uM of remdesivir. However, the same mutant virus showed constant in titer of 10^1.5 at 24hr and 48hr under 2uM of remdesivir. Could the authors explain why the mutant virus grows inferior under low concentration of RDV? Could the titer of the mutant virus increase when culture 72 hr under 1uM of RDV?

Reviewer #2: In vitro Remdesivir resistance was acquired rapidly. The mutations described here are not frequent in SARS-CoV-2 cases. They also do not emerge in documented long-term infections of patients who underwent Remdesivir treatment. The authors should discuss this disconnect.

The discussion of the spike mutations requires more nuance. The acquired mutations that confer antibody resistance are at sites that tolerate mutation. Selection by antibodies can result in selection for variants at these positions. The authors have identified sites that can mutate when viruses are passaged extensively. Most discussion surrounding substitutions at 501 and 655 has not been centered on antigenicity.

**Part III – Minor Issues: Editorial and Data Presentation Modifications**

Reviewer #1: 1. Line 105-108, EC50 needs to be consistent since IC50 was used in the figure legend

2. Line 127, should be “result in” instead of “reduce in”

3. The naming system should be consistent, such as using SARS-CoV-2wu1 in Line 120 but using rSARS-CoV-2 in line 143 or 151….

4. Figure 3 should be more self-explanatory by showing the titers tested in which cell lines since 3B was done in vero-EMPRSS and 3C was done in calu-3.

Reviewer #2: Unfortunately since the time of this manuscript’s drafting over a million more deaths due to SARS-CoV-2 have occurred. Before publication please update this information.

Line 132: Use of highly conserved. Things are either conserved or not and do not need a qualifier.

I found the “Prevalence of RDV mutations in circulating populations” section difficult to follow. Please consider reworking it.

PLOS authors have the option to publish the peer review history of their article (what does this mean?). If published, this will include your full peer review and any attached files.

Reviewer #1: **Yes: **Day-Yu Chao

Reviewer #2: No
---

## [Decision Letter · Decision Letter 1]

30 Aug 2021

Dear Dr Stewart,

We are pleased to inform you that your manuscript 'In vitro selection of Remdesivir resistance suggests evolutionary predictability of SARS-CoV-2' has been provisionally accepted for publication in PLOS Pathogens.

Best regards,

Shin-Ru Shih

Section Editor

PLOS Pathogens

Shin-Ru Shih

Section Editor

PLOS Pathogens

Kasturi Haldar

Editor-in-Chief

PLOS Pathogens

orcid.org/0000-0001-5065-158X

Michael Malim

Editor-in-Chief

PLOS Pathogens

orcid.org/0000-0002-7699-2064

Reviewer Comments (if any, and for reference):

Reviewer's Responses to Questions

**Part I - Summary**

Reviewer #1: The authors have properly answered the questions and revised the manuscript based on the comments.

Reviewer #2: The authors have improved this manuscript and have addressed my concerns. I found a few very minor suggestions to improve the presentation of this work.

**Part II – Major Issues: Key Experiments Required for Acceptance**

Reviewer #1: none

Reviewer #2: N/A

**Part III – Minor Issues: Editorial and Data Presentation Modifications**

Reviewer #1: none

Reviewer #2: Minor:

Figure 1. legend space before panel “c”

Figure 1: “Virus stock” or “stock” or “parental” may be more informative than the column labeled SARS-CoV-2 in panels D, E and F. This is entirely at the authors discretion.

Line 189: “did not share homology” may not be the right term. Perhaps “suggesting that they arose independently”

Line 354: Lines of the text are partly duplicated

PLOS authors have the option to publish the peer review history of their article (what does this mean?). If published, this will include your full peer review and any attached files.

Reviewer #1: **Yes: **Day-Yu Chao

Reviewer #2: No

---

## [Editor Report · Acceptance letter]

13 Sep 2021

Dear Dr Stewart,

We are delighted to inform you that your manuscript, "*In vitro* selection of Remdesivir resistance suggests evolutionary predictability of SARS-CoV-2," has been formally accepted for publication in PLOS Pathogens.

Best regards,

Kasturi Haldar

Editor-in-Chief

PLOS Pathogens

orcid.org/0000-0001-5065-158X

Michael Malim

Editor-in-Chief

PLOS Pathogens

orcid.org/0000-0002-7699-2064